# How Different Tools Contribute to Climate Change Mitigation in a Circular Building Environment?—A Systematic Literature Review

**Lucas Rosse Caldas** [1,2,*] **, Maykon Vieira Silva** [3] **, Vítor Pereira Silva** [3] **, Michele Tereza Marques Carvalho** [3] **and Romildo Dias Toledo Filho** [1]

1. Programa de Pós-Graduação em Engenharia Civil, PEC, COPPE, Universidade Federal do Rio de Janeiro, Cidade Universitária, Rio de Janeiro 21945-000, Brazil; toledo@coc.ufrj.br
2. Programa de Pós-Graduação em Arquitetura, PROARQ, FAU, Universidade Federal do Rio de Janeiro, Cidade Universitária, Rio de Janeiro 21945-000, Brazil
3. Programa de Pós-Graduação em Estruturas e Construção Civil, PECC, Universidade de Brasília, Campus Darcy Ribeiro, Brasília 70910-900, Brazil; eng.mayconsilva@gmail.com (M.V.S.); victorpereira._14@hotmail.com (V.P.S.); micheletereza@gmail.com (M.T.M.C.)
* Correspondence: lucas.caldas@fau.ufrj.br; Tel.: +55-62-9672-7202

**Abstract:** The circular economy (CE) has become a trend because concern has arisen regarding the end of life of several products and the reduction of $CO_2$ emissions in many processes. Since the architecture, engineering, and construction (AEC) industry is one of the biggest generators of environmental impacts, there is a need to apply the CE concept to the industry in order to reduce greenhouse gas (GHG) emissions. However, the role of different tools that are used to integrate CE strategies to reduce GHG emissions by the AEC industry is still unknown in the scientific literature. The purpose of this paper is to carry out a systematic literature review on the theme and analyze the following seven tools: (1) life cycle assessment—LCA; (2) building information modeling—BIM; (3) building environmental certifications—BEC; (4) building materials passports—BMP; (5) waste management plan—WMP; (6) augmented reality—AR; and (7) virtual reality—VR. A total of 30 papers were reviewed, and it was observed that, in terms of CE strategies and climate change mitigation, the vast majority can be classified as closing loops and are mainly related to recycling and reuse at the end of life and the use of recycled materials. Considering the building's stakeholders, constructors, researchers, and designers can be the main users and, consequently, those that most benefit from the use of the evaluated tools. The integration between LCA, BIM, and BMP was also observed. Finally, as one of the main contributions of this research, other types of integration among the analyzed tools are proposed. These proposals seek to improve and update the tools and also address the need to reduce GHG emissions.

**Keywords:** circular economy; buildings; life cycle assessment; building information modeling; materials passport; virtual reality

## 1. Introduction

The building sector is one of the major contributors to greenhouse gas (GHG) emissions, depletion of natural resources, and waste generation [1,2]. In this perspective, it is necessary to change the way cities, buildings, and their various elements are designed. For this, it is necessary to change the current linear way of thinking to a circular model, in which resource use efficiency is increased and waste and pollutant generation is reduced. With this vision, it is possible to make cities and buildings more inclusive and sustainable.

The circular economy (CE) model has gained attention in recent years from several productive sectors, including the architecture, engineering, and construction (AEC) industry [3,4]. There are different principles or strategies for implementing a CE model,

including use of waste from other processes, reduction of natural resource consumption, prioritization of the use of renewable resources, deconstruction or design for disassembly project (DfD), design for performance, design for service life extension, construction virtualization, end-of-life reuse and recycling, etc. [5–7].

When applied to the AEC industry, the CE model should encompass the entire life cycle of a product. It can include a material, a piece of furniture, a construction element (wall, floor, roof, etc.), or an entire building [3]. In addition, it is known that the construction sector is composed of several actors with different roles, such as developers, builders, designers, materials suppliers, building users, managers, etc. Thus, it is clear that the study of the CE concept applied to this sector tends to be complex, and some opportunities may not be fully explored.

Most studies in the literature regarding the CE applied to the construction sector have focused on the use of waste for materials production and reuse strategies, and many other strategies have not yet been explored in depth. In addition, few studies have focused on carrying out a quantitative analysis of the literature on the CE and AEC industry. Some of the research, such as that of Akanbi et al. [8], Akanbi et al. [9], and Honic et al. [10], has used tools, such as life cycle assessment (LCA), building information modeling (BIM), and materials passports (MP), respectively, to facilitate the production process for circular construction products. However, it is still a research topic that needs to be further explored.

New research should include the evaluation of other tools that can help the implementation of CE strategies. Some strategies, such as waste management plans during buildings construction and buildings environmental certification schemes, e.g., leadership in energy and environmental design (LEED) or building research establishment environmental assessment method (BREEAM), have been used in the construction sector for some time. Recently, with the increase in interest in smart building development, the use of information and communication technology (ICT) has aroused great interest in research [11]. Therefore, augmented reality (AR) and virtual reality (VR) can be considered potential tools for a more sustainable building design development since they can lead to the virtualization of the construction sector.

In the literature, some review studies have already been published regarding the CE in the AEC industry. Gallego-Schmid et al. [12] evaluated the links between the use of CE strategies for climate change mitigation. López Ruiz et al. [13] reviewed different studies related to the CE and waste generated in construction and demolition activities. Foster [3] evaluated the use of CE strategies with a special interest in historic buildings. Hossain et al. [14] performed a systematic literature review to evaluate the implications, considerations, contributions, and challenges of the CE in the construction industry. These authors identified the existing trends and challenges in different parts of the process (design, materials selection, supply chain, business model, risk management, etc.) and actors. They observed that just a small percentage of studies focused on the environmental assessment and, when it is performed, the LCA is the most used tool. Ávila-Gutiérrez et al. [15] developed a framework aligned with the goals of the 2030 Agenda for Sustainable Development under the three pillars of sustainability and industry 4.0. Superti et al. [16] developed a framework for the construction and demolition sector that categorizes CE interventions into four parts: research and realize, implement, support, and enable, each considering the so-called 10R-strategies commonly used in the CE universe. Ogunmakinde et al. [17] assessed the link between the CE and the United Nations Sustainable Development Goals (SDGs). They observed that is essential to understand the relationship between CE strategies and the SDG in order to attain smarter construction and demolition waste management and that all stakeholders who generate waste have an important role in the transition to a circular model. Norouzi et al. [18] performed a quantitative scientific evolution analysis of the application of CE in the construction sector by the analysis of 7000 documents published between 2005 to 2020. They verified that researchers pay close attention to the following areas: sustainability, energy efficiency, renewable energy, LCA, and recycling.

However, there is still a knowledge gap in research focused on the use of different design, management, and execution tools for the production of more circular buildings that, at the same time, can reduce their GHG emissions. This gap is even greater when considering the seven tools selected in this research. It is possible to observe that most studies present frameworks linked to CE strategies but most of them do not offer ways to implement them considering the tools available in the sector.

Based on this context, we propose the following research question: "How can the use of different tools contribute to the implementation of CE strategies and simultaneously attain the reduction of GHG emissions?" Therefore, the objective of this study is to evaluate via the scientific literature how different tools used in the AEC industry can contribute to the mitigation of climate change in a CE environment with a focus on building life cycles and stakeholders.

This work also aims to contribute to the acceleration of actions to mitigate GHG emissions through the CE and, thus, to the SDGs advocated by the UN 2030 Agenda, more specifically for SDG 13—actions against global climate change. In addition, the implementation of the CE is considered to have a direct impact on SDG 12—sustainable production and consumption—and SDG 11—making cities and communities inclusive, safe, resilient, and sustainable. For this reason, the CE is considered essential in achieving sustainable development [19].

The following tools were chosen in our research: (1) life cycle assessment (LCA), (2) building information modeling (BIM), (3) building environmental certifications (BEC), (4) building materials passports (BMP), (5) waste management plans (WMP), (6) augmented reality (AR), and (7) virtual reality (VR). These tools were selected based on previous research and the construction practices related to the theme of sustainability of the construction sector and more recently with the CE [20]. The last two tools, AR and VR, were added because they are starting to receive attention in the context of the circular economy and digitalization of the built environment [15,21].

To our knowledge, this is the first research that evaluated, via a systematic literature review, the study of these tools in order to apply almost 20 CE strategies in the building sector. Many measures, such as waste management and lean construction, have already been taken to reduce carbon emissions by the AEC industry [22]. Minimizing these emissions brings improvements in management and design concepts and the development of new technologies in the sector linked to CE strategies [23]. Therefore, this research seeks tools and technologies that contribute to the development of the CE in the AEC industry, addressing their interactions at all stages, from the design phase to demolition or deconstruction, with the primary purpose of combating climate change.

As its main contribution, this study presents the state-of-art of what has been done in different research centers worldwide and the main existing knowledge gaps that must be overcome. Additionally, we propose a type of classification for the use of tools, items that the evaluated tools can improve for better integration between them, which will allow a more circular process and at the same time contribute to climate change mitigation. We also evaluated which are the main stakeholders of the AEC industry that would benefit from the use of these tools. This research should serve as a guide for other researchers, designers, builders, and all AEC industry stakeholders who strive to understand and implement these tools in design development with CE principles during a building's life cycle.

## 2. Methodology

This paper is divided into three main parts. First, a quantitative analysis is performed to identify the most studied tools and the ones most related to the mitigation of climate change (in terms of GHG emissions reduction). Second, a qualitative assessment was carried out to understand the benefits and obstacles of each tool in the implementation of the CE, and proposals are presented for reducing climate change, considering the different strategies, stakeholders, the building's life cycle, and the integration between them. Finally, recommendations for the improvement of tools are made.

The methodology involved a systematic literature review (SLR) that is based on the study of Charlotte et al. [24]. These authors proposed design and construction strategies for the CE applied in the AEC industry. However, they did not focus on the understanding of how different tools can be used or on climate change mitigation targets. On the other hand, our SLR focuses on identifying and recording the main points related to climate change mitigation and the CE using any of the seven proposed tools.

The Scopus and Web of Science databases were used for literature searches using a search string with a specific set of predefined keywords: "circular economy" AND "climate change" AND "construction". The period was restricted to publications from the last 5 years.

The purpose of the literature review is to identify the current state-of-the-art related to the proposed theme. From this, it is possible to carry out quantitative and qualitative analyses and provide new contributions that promote the evolution of science in the civil construction sector, mainly based on the CE, which needs to be thoroughly consolidated in the AEC industry. The following inclusion criteria for the research were applied:

- Publications should be related to the area of the AEC industry.
- Publications should present explicit strategies related to the concept of the CE.
- Strategies should focus on actions that seek to mitigate global climate change.
- Publications should also address one or more tools proposed for use.

In the searches of the two databases, a total of 79 publications were found. After excluding any duplicated publications and making a first analysis based on the title, abstract, and keywords, this number was reduced to 33. At this stage, papers in which the CE was not the main topic of the research, or those in which some of the tools or actions to mitigate climate change were not present, were discarded. Following this, reading of the introduction and conclusion was carried out. In all, from the Scopus database, 14 papers were selected as relevant and of these, 6 were included in the synthesis. From the Web of Science database, 19 papers were initially selected and 6 were finally included.

The snowball approach added 18 new papers to the analysis, with some of them being previously known to the authors, which gave a final total of 30 publications. The outline of the selection method is shown in the flowchart in Figure 1.

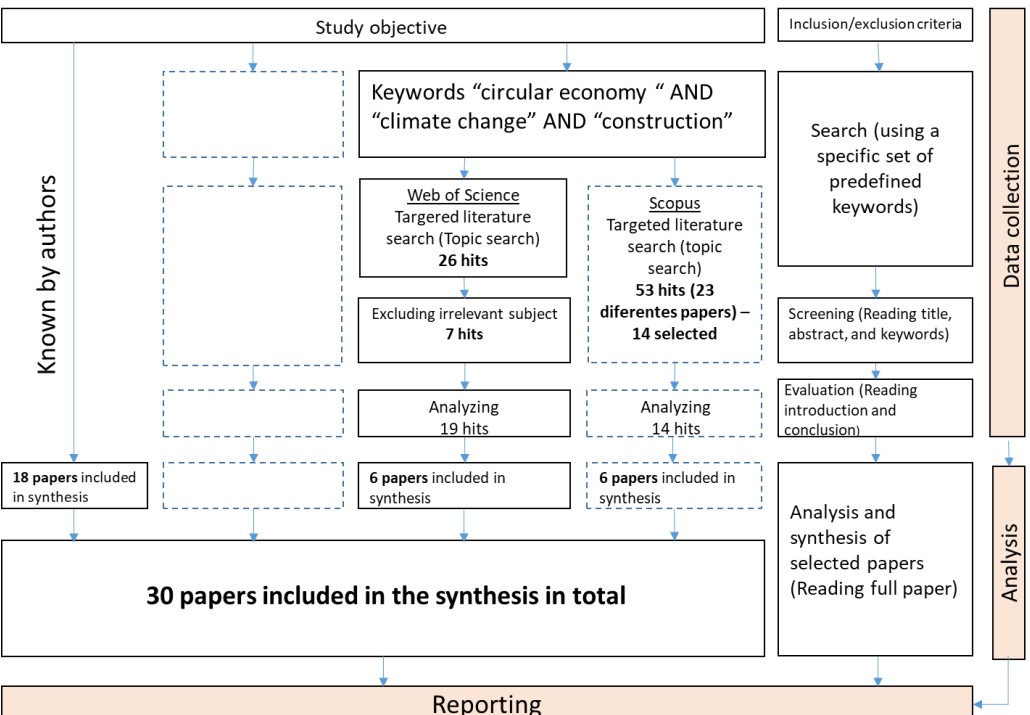

**Figure 1.** Flowchart with a summary of the selection method.

In this study, the following tools were evaluated based on BAMB (2020):

(1)   life cycle assessment (LCA);
(2)   building information modeling (BIM);
(3)   building environmental certifications (BEC);
(4)   building materials passport (BMP);
(5)   waste management plan (WMP);
(6)   augmented reality (AR); and
(7)   virtual reality (VR).

The definition of the CE strategies to be evaluated for the construction sector was carried out based on the literature review, which was guided mainly by the following research: EMF [5,6], Malmqvist et al. [25], Cheshire [26], and Bocken et al. [27].

The CE model can be defined as "a regenerative system in which the entry of resources and waste, emission and energy losses are minimized by the deceleration, closing and narrowing of the material and energy loops [28].

Based on research by Bocken et al. [27], this paper also uses classification in terminologies to categorize CE strategies according to the mechanisms through which resources flow through a system. There are 3 categories (groups):

(1)   slowing loops
(2)   closing loops
(3)   narrowing loops

Each group has a series of CE solutions that demonstrate how each strategy can be implemented in practice. The description of this categorization can be seen in Table 1.

**Table 1.** CE strategies evaluated in the study.

| Classification | Strategies |
|---|---|
| Closing loops | (1) Use of reused and recycled materials; (2) reuse of buildings; (3) design for deconstruction; (4) use of renewable sources; (5) reuse at the end of life; (6) recycling at the end of life. |
| Slowing loops | (1) Design for low maintenance of the construction and its components; (2) design for performance; (3) design for service life extension of the construction and its components; (4) design for adaptability of the construction and its components; (5) design for multifunctional products. |
| Narrowing loops | (1) Optimization of floor (usable) area; (2) optimization of energy and water use; (3) sharing spaces; (4) use of industrialized and lightweight building systems; (5) construction virtualization. |

## 3. Quantitative Analysis

In this section, we present the results quantitatively, first considering a preliminary analysis and then based on the analyzed literature.

### 3.1. Preliminary Analysis

Before starting the research that focused on measures to mitigate climate change, an investigation was carried out to evaluate the current research scenario related to the CE. This exposed the period of greatest research development, the main countries per number of contributions, and the use of the tools in civil engineering.

The concern about the CE as an academic issue is recent. From 2015 to the present, 6874 results were found on the Web of Science when the keyword "circular economy" was used. The majority of the documents found are related to other areas, which shows a knowledge gap regarding this specific topic and the construction industry. Almost 40% of them were published in 2020. When refined by using the keywords "civil engineering" and "architecture", the result was 194. The countries with the most research done so far are Italy, England, and Poland with 15%, 9%, and 8%, respectively.

In this paper, a scientific metric literature review was conducted using VOSViewer and Scopus to create a database containing 484 papers to identify the main research keywords from literature published between 2016 and 2021 and related to the CE in the construction industry. An overlay visualization map (Figure 2) was produced using the most frequently used keywords in the literature, and climate change, sustainable development, and circular economy represented the three main clusters. The size of the circles represents the occurrence of each keyword in the papers. The first (in blue) is related to climate change and gases that contribute to pollution, the second (in green) is linked to the CE and sustainability, and the third (in red) presents the most common strategies and tools. The 16 keywords that are related to CE strategies and/or tools were gas emissions, climate change mitigation, greenhouse gases, carbon dioxide, climate change, CE, sustainable development, biomass, economics, sustainability, economics, recycling, life cycle assessment, waste management, article, environmental impacts, and environmental management.

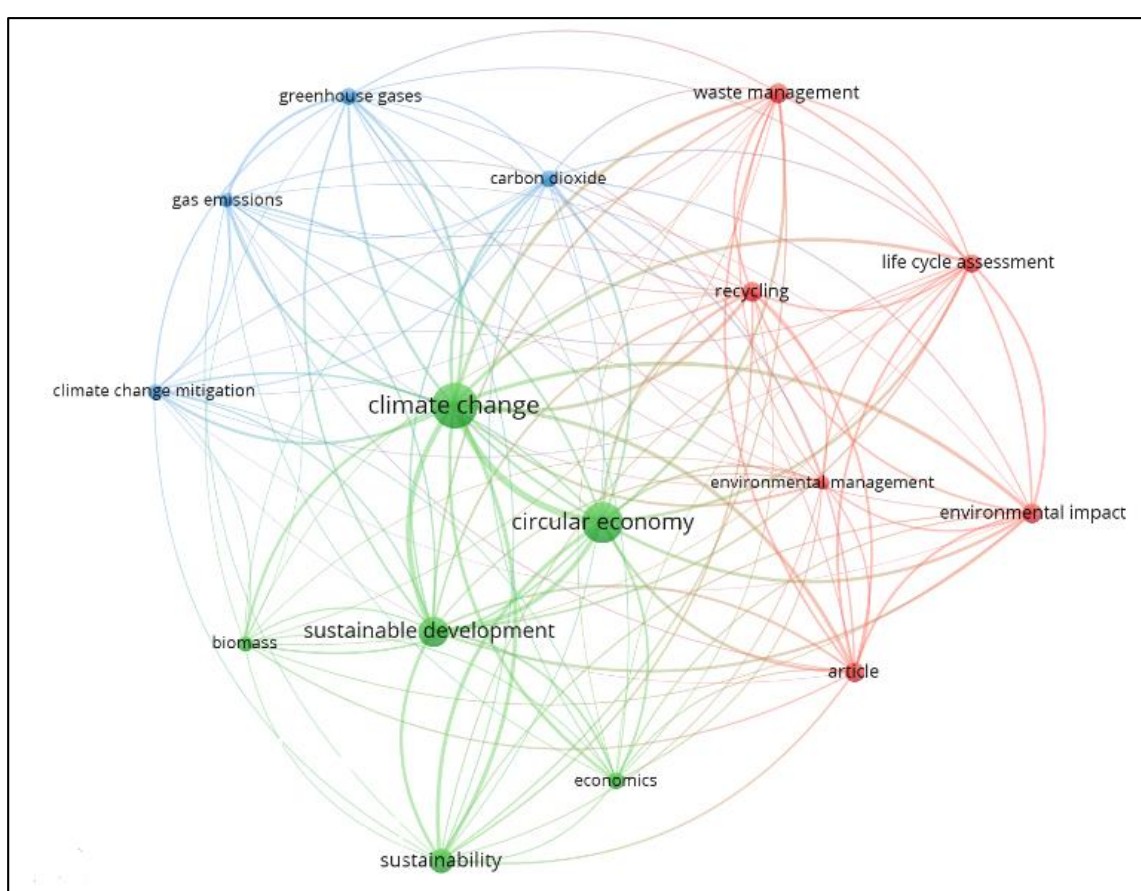

**Figure 2.** Overlay visualization map.

When searching for the keywords "circular economy" and "construction" on the Scopus database, we can see how the theme has become more relevant from 1985 to the present day. The concern regarding the service life of the construction and its components is among the most relevant guidelines for achieving sustainability in this sector. Figure 3 shows expressive growth since 2009, and the graph shows that CE research gained more prominence after 2015 and that, since then, there has been a sharp increase in the number of works published.

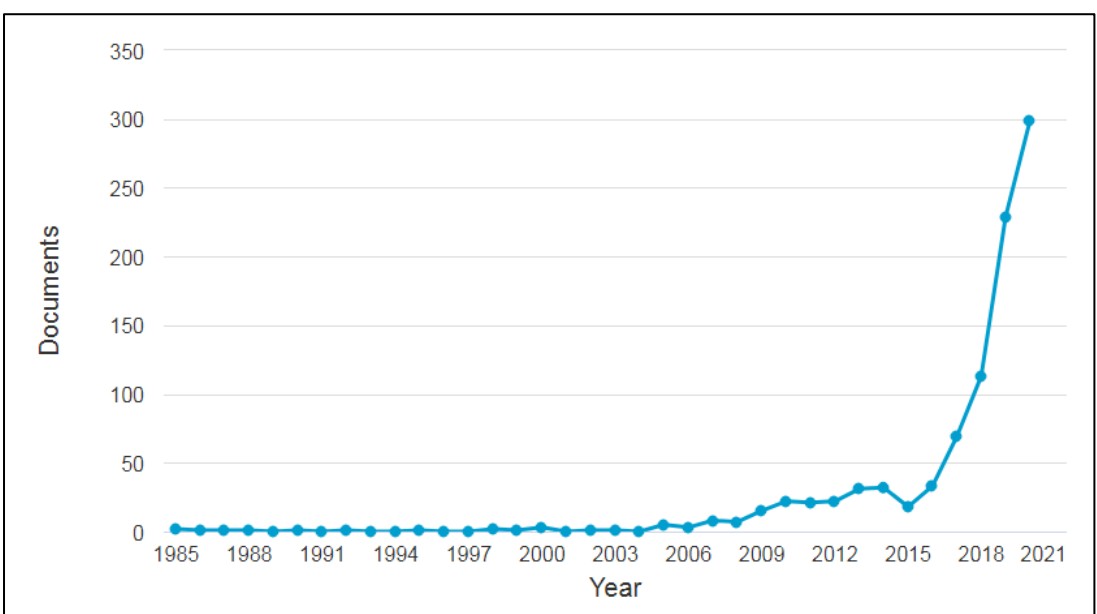

**Figure 3.** Number of papers published each year.

This expressive amount of papers after 2012, starting in 2015, can be related to the first and very well accepted publication, especially by the market and governments, of the CE, which was published by the Ellen McArthur Foundation [5] in 2012. In 2014 and 2015, the European Commission (EC) strategy was outlined and revised, respectively [18]. In 2015, we have the Paris Agreement with the definition of the goals to reduce GHG emissions in a lot of countries and the publication of the 17 SDGs by the United Nations. After 2018, the concept of the CE became widespread around world including developing countries, especially China, where a great number of publications of this theme came from, and in the AEC industry and construction sector. In the last five years, research linking the CE and climate change mitigation and SDGs already started to be more frequent [29,30].

Using the same keywords, we obtained a list of the countries that have published the most on the topics during 1985–2020. China stands out with 229 publications, followed by Italy with 97 publications and Spain with 87 publications (Figure 4). Brazil appears in 20th place with 14 publications.

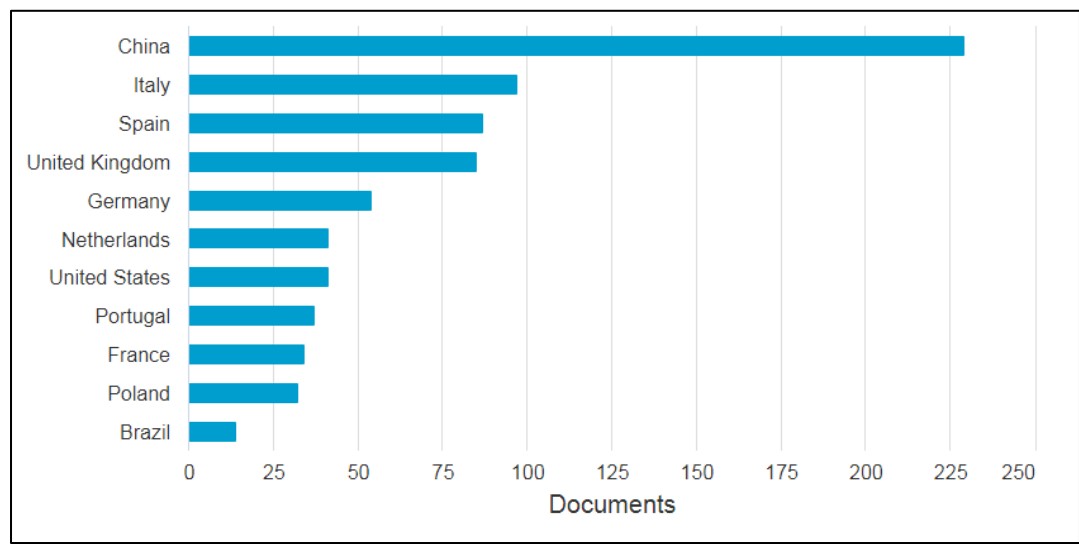

**Figure 4.** Countries with the greatest number of publications with the circular economy as their theme.

Two databases were used to accomplish the SLR: Scopus and the Web of Science. A comparison was made between them using keywords related to the respective tool, in the areas of civil engineering and architecture (when associated with the Web of Science) and in the "engineering" area with the additional keyword "building" or "construction". The percentage of publications found related each tool is presented in Figure 5. It can be seen that the participation of some tools, such as BIM and LCA, was much more often explored, with a total of 74% on the two themes (including duplicate articles). However, the other construction tools were much less often researched, evidencing a gap in the area that can be further explored, especially when it comes to BEC, BMP, and WMP.

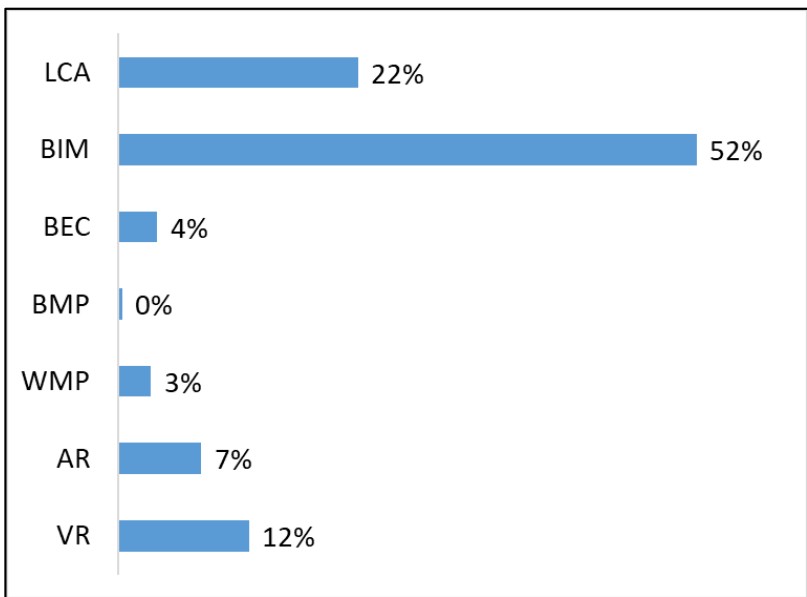

**Figure 5.** Research addressing tools in civil engineering.

*3.2. Use of Tools Considering the Context of Circular Economy and Mitigation of Climate Change*

An analysis of the main categories of the CE was made based on the 30 articles that met all the content requirements in the methodology. It was noted that 68.8% of the papers have aspects of closing loops as their main theme, 13.3% are related to slowing loops, and 18% focus on optimized flow (Figure 6). The details of the CE strategies evaluated are presented in Figure 7. It is important to note that one paper can have more than one aspect/strategy of CE, and can be classified as closing loops, slowing loops, or optimized flow.

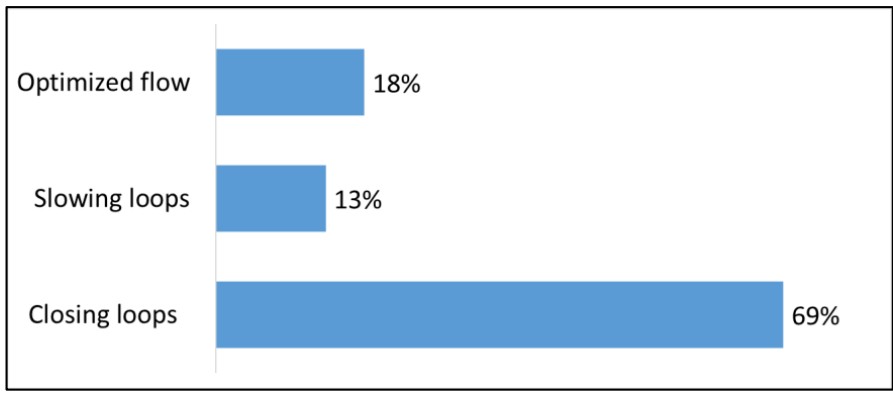

**Figure 6.** Classification of the categories of the strategies in the circular economy.

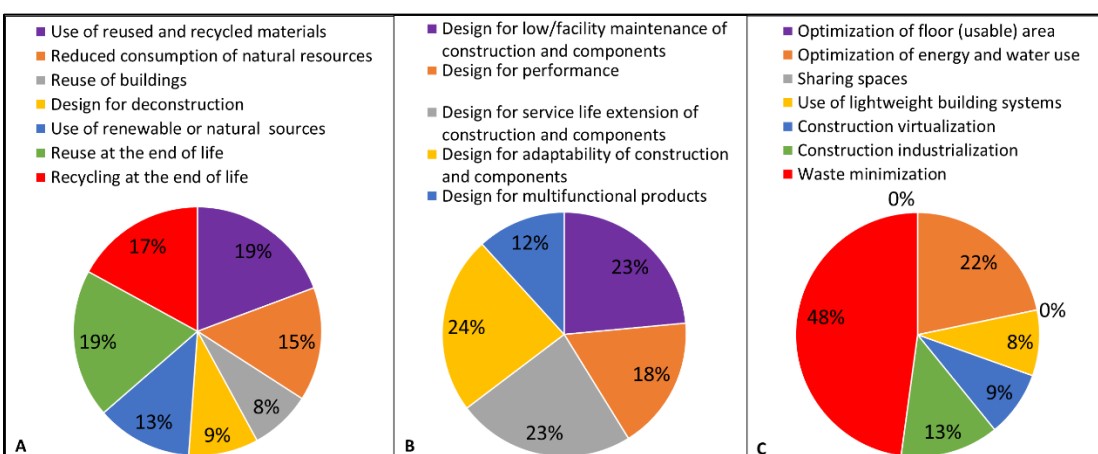

**Figure 7.** Categories of the strategies in the circular economy for each classification. (**A**) Closing loops, (**B**) slowing loops, and (**C**) narrowing loops.

When dealing with aspects of the CE addressed in the papers, the vast majority have strategies related to closing loops (69%), and the rest are mainly related to recycling (17%) and reuse at the end of life (19%) and the use of recycled materials (19%). Within this category, there are also papers on the reduction in the consumption of natural and virgin resources (15%) and, in some cases, on the use of renewable or natural resources (13%). There is limited research that explores the reuse of buildings (8%) and design for deconstruction (9%).

Strategies for slowing loops represented just 13% and the desegregated data show that design for adaptability (24%), design for service life extension (23%), and design for low maintenance (23%) were the most studied. Finally, the strategies for narrowing loops represented 18%, and their focus was on waste minimization (48%) and optimization of energy and water use (22%). By evaluating the three groups of CE strategies, we can see that most of the attention of the researched literature has been given to issues related to waste.

The closing loops strategies are strategies that will return the generated waste to the beginning of process (the same one or another one), which is aligned with the cradle-to-cradle principles. This is directly related to reuse and recycling and the decrease in consumption of natural resources. Most of the reviewed literature focuses on this kind of strategy. One possible explanation for this is that most of the literature focuses on resources at the material level, which is easier to be evaluated than a whole building.

The slowing loops strategies are strategies that are concerned with the extension of the service life of the product (which can be a single type of material or the whole building). Therefore, the increase in the number of years that the product maintains its function for the user is the main goal. Finally, the narrowing loops strategies are strategies that seek resource efficiency and have the objective of using fewer resources per product [27]. At the building level, one of the most efficient strategies is the reduction of built area and the reduction of energy and water use. Virtualization is another very promising strategy since it could replace physical consumption with virtual consumption.

When analyzing the number of studies that use the tools to develop actions that contribute to climate change mitigation in a CE environment, Figure 8 indicates that LCA is the most used, followed by BIM. In all, 67% of the total sampling of occurrences of any tool is represented by LCA, while for BIM, this value is 16%. The least used tools are information and communication technology (ICT), AR and VR, and BEC, all of which are below 10% It is important to highlight that the vast majority of the works (about 80%) address a building material in their studies, concrete being the most mentioned, which appears in 35% of the works.

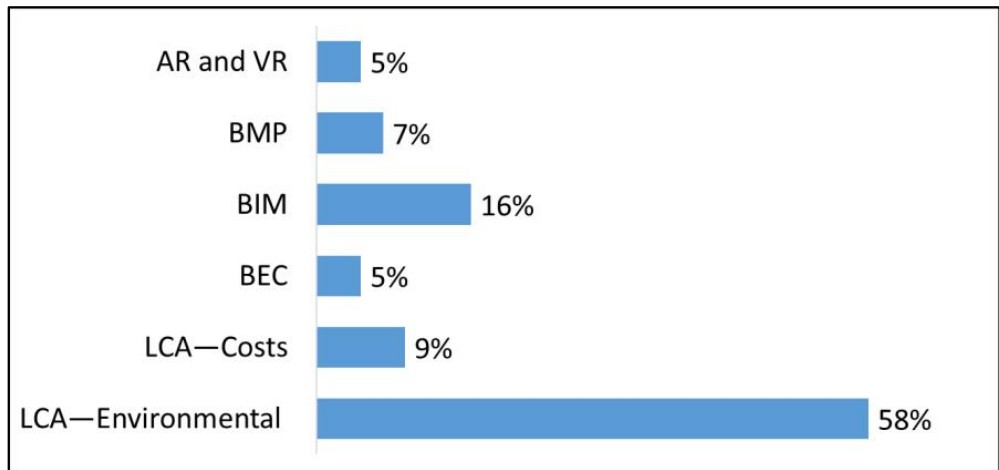

**Figure 8.** Quantitative analysis of the literature regarding the use of tools to enhance the circular economy and mitigate climate change.

Another approach, analyzed in Figure 9, is the stage in which the use of the CE strategies is proposed. In general, there are studies of the tools for all stages of the building's life cycle; however, the techniques are more often explored in the construction (36%) and the end of life (31%) phases. From the point of view of the effectiveness of the adoption of CE principles, we know that the design stage is the most important [31] since it defines most of the strategies and solutions at a lower cost. The end of life stage presented a significant share, which is related to most strategies adopted and is related to reuse and recycling, since these are classified as closing loops.

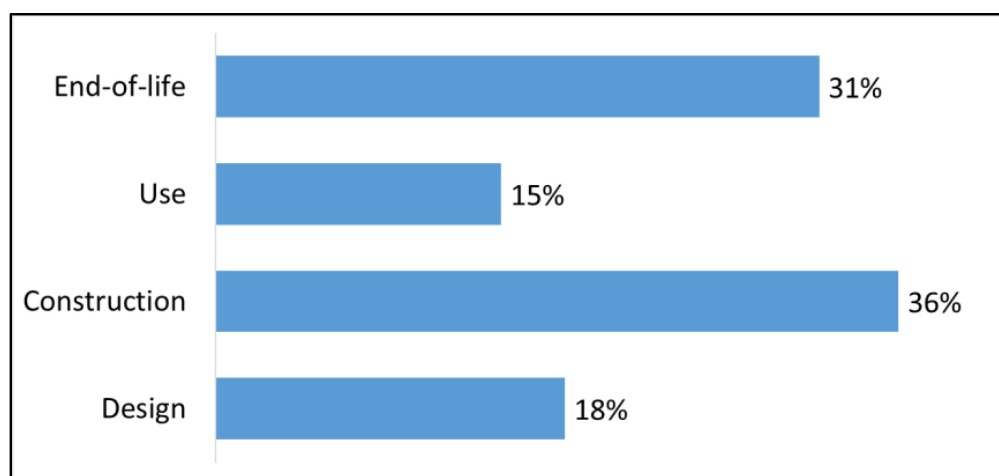

**Figure 9.** Lifetime stage in which circular economy strategies are applied.

Finally, Figure 10 presents a survey of climate change and other environmental aspects/impacts addressed in this work. It can be seen that all the papers address climate change and that 32% relate it to energy consumption. The link between climate change and energy is expected, as an important part of GHG emissions comes from energy consumption (especially fossil fuels).

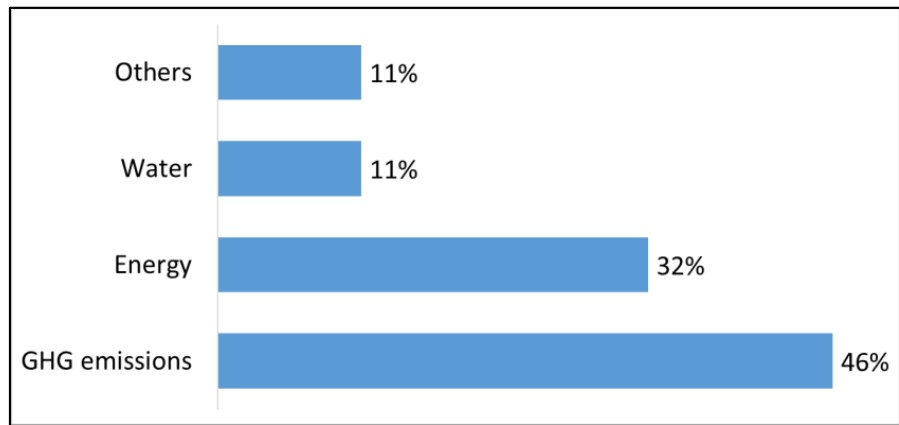

**Figure 10.** Environmental aspects/impacts evaluated in the literature regarding the use of tools to enhance the circular economy in the building environment.

## 4. Qualitative Analysis

In this section, we present each of the tools evaluated in more detail. These are evaluated qualitatively, based on the analyzed literature, and emphasize each one's benefits and obstacles/difficulties.

### 4.1. Life Cycle Assessment (LCA)

The main contribution of the use of LCA is to account for the benefits of the CE in the construction sector, and these can be summarized here: (1) the quantification of the benefits due to avoided impacts (of reused and recycled materials); (2) the quantification of biogenic carbon when bio-based materials are used (in terms of climate change impact); (3) comparison of building products with and without the use of CE principles; (4) comparison between end of life options.

According to EN 15978 [32], module D is destined for the quantification of "Benefits and Loads Beyond the System Boundary—Reuse, Recovery, Recycle Potential". Some studies focused on the evaluation of this module, such as Anderson et al. [33] and Rasmussen et al. [34]. The product environmental footprint (PEF) has been recently developed by the European Commission as a common method for the assessment of the environmental performance of products. While the end of life stage is not a mandatory stage for EN 15804 [35], for PEF, it is considered in the system boundaries and life cycle stages, including product recovery or recycling [36]. Therefore, the use of PEF can facilitate the evaluation of CE strategies, especially in terms of the influence of end of life strategies.

In the case of the use of bio-based material, the quantifying of benefits due to biogenic carbon for climate change impact has a great influence on the LCA's results. Although there is notable divergence in the literature in relation to methodologies [37–39], studies tend to calculate the biogenic carbon, especially in building products that have a longer service life, and consider dynamic aspects [40,41].

The integration of LCA and BIM also shows good potential for facilitating the evaluation of CE strategies for the development of building products, especially in terms of the automation process of LCA. Some points in LCA studies must be documented and be transparent in terms of the benefits of the CE: reuse/recycling rate and transportation distances. These factors can have an important influence on LCA results, as pointed out by Cruz Rios et al. [31] who compared the reuse of steel and wooden exterior wall framing systems.

Most studies used the attributional approach, which, from the point of view of avoided impacts, can lead to less assertive and systematic conclusions. However, we expect that with the evolution of consequential approaches and data, this will change in the future.

In terms of dimensions of sustainability, the environmental dimension is the most used. Some studies evaluated the economic aspect [42,43], and none evaluated the so-

cial dimension. For the environment evaluation, climate change impact was the most often assessed.

Niu et al. [44] carried out a case study involving the LCA of wooden sheds, which considered the concepts of the CE, in order to quantify the potential to combat climate change by reusing wood at the project level. The results showed positive impacts with the reuse of wood, especially in terms of $CO_2$ emissions and the reduced use of raw materials.

Mostert et al. [45] used the LCA as an environmental assessment tool for urban mining in the construction sector to investigate different recycled concrete scenarios in order to assess the best environmental performance. The objective of the LCA was to quantify the material, energy, water, and carbon footprint. The method proved to be suitable for environmental assessment, considering a CE approach.

In terms of loop deceleration, some researchers have focused on design for disassembly (DfD) as a key solution to facilitate material reuse, including method development in order to quantify the potential for avoiding GHG emissions. For example, the work of Eberhardt et al. [46] proposed an LCA method for quantifying the reduction of emissions and the potential environmental savings from the application of DfD for concrete structures in order to optimize combinations of material choices, extend the life of buildings, and facilitate the reuse of building materials.

### 4.2. Building Information Modeling (BIM)

Depending on where it is applied, during the building's life cycle, BIM can be used as a process and an assessment tool to facilitate the implementation of the CE in the construction sector. In the early design stages, it can be used for automatic quantitative extraction and can be coupled with BMP and LCA to provide the environmental and recyclability potential of building materials and components [10,47]. BIM can be used to help in the design of buildings according to environmental certification, e.g., LEED [48,49] or LCA [50,51]. During the building's construction, it can be used to optimize the use of energy and water and reduce waste generation. In the use stage, it can be used as a management tool to aid in maintenance planning and provide more efficient management of the resources (energy, water, etc.) used in the operation of the building [52]. Finally, in the end of life stage, it can help waste management, especially in terms of quantification of generated waste and the evaluation of the recycling potential of some of the materials [8].

Recently, special interest has been paid to the use of BIM to facilitate DfD. Akinade et al. [53] established a score for evaluating the level of building deconstruction of a building's design based on a BIM deconstruction assessment framework. Akanbi et al. [8] enumerated the essential functionalities required of the BIM-based deconstruction process. Akanbi et al. [9] created a deconstruction and disassembly system that permits designers to assess different end of life performance options in the early design stages and, if necessary, make some adjustments. The system enables the increase in material reuse in a building design and in reduction of waste generation and extraction of virgin raw materials.

The actual process of quantifying GHG emissions during the product materialization stage is difficult and complex, as different types of materials, machines, and construction technologies are densely mixed in a short period. Hao et al. [23] developed a BIM-based approach for quantifying and reducing GHG emissions in prefabricated construction projects. They found that BIM is an efficient tool for measuring GHG emissions and, with its use, they observed reductions of up to 15% in GHG emissions using prefabricated methods compared to the traditional ones.

BIM offers an advanced tool for sustainability analysis for construction and projects because it has the advantages of visualization, coordination, simulation, and optimization [54,55]. Abanda and Byers [56] applied BIM to investigate the impacts of the building's orientation on energy consumption and optimization of design alternatives. Röck et al. [2] proposed an advanced method by applying a visual script to connect LCA and BIM to allow designers to identify and view specific project access points with the potential to reduce the building's environmental impact. These studies illustrate that BIM can be used to simplify the process

of analyzing GHG emissions for construction projects, and thus has the benefit of reducing environmental impacts such as those involving climate change.

BIM is a powerful tool that helps obtain material and energy consumption data in the construction material production and transportation process, and this is possible through the list of materials that BIM can offer. BIM can become a database containing rich engineering information on materials and components and can provide reliable information on resource consumption [57]. According to this information, quantities can be obtained automatically and precisely, thus reducing manual operations and the associated errors in the quantity surveying process. These more accurate data will provide greater reliability in the LCA. In this context, embedding environmental parameters and calculations directly into the model in order to extract calculated impacts instead of pure material quantities (BIM4LCA) can improve system-wide performance [58].

### 4.3. Building Materials Passport (BMP)

The BMP can be considered a tool for facilitating the implementation of the CE in the early design stages. It delivers detailed data and information about the quantitative and qualitative composition of materials, components, or building elements [47]. The concept and diffusion of the BMP gained strength during the project "Buildings as Material Banks" (BAMB) funded by the EU, and pretends to increase the value of building materials, reduce waste, and use fewer virgin resources ("BAMB", 2020). However, there are just a few examples in literature. Honic et al. [10] used a BMP framework integrated with BIM and LCA to compare two constructive systems, in this case, concrete and wood, during the end of life of a residential building in Austria. They found that the recycling potential of the concrete alternative is better, while the wood option leads to less waste. The authors emphasize the large potential of the BMP, especially when integrated with BIM and LCA, since it improves the recyclability of buildings (new and existing ones).

Munaro et al. [47] created a framework for the development of the BMP during different stages of the building's life cycle, and showed the data and responsibilities needed in each stage. They applied the proposed BMP in a wood-framed building and observed some difficulties. Atta [59] developed a framework based on the BMP, LCA, and BIM that pretends to automate the sustainability evaluation by using different indicators and by facilitating the documentation and sharing of the building's information for future needs. They applied the framework by comparing a traditional residential building with new alternatives of modular building. They observed that modular building is more sustainable and is more compatible with the CE concept.

In addition, the role of companies in seeking to reinforce their responsibility as producers and in rethinking their business models is highlighted. In this way, they have to take into account the entire product life cycle, from extraction to disposal, and take actions related to product design strategies, such as the choice of material inputs in the production process that will allow it to be repaired, reused, or recycled later on, as well as respect for the product's service life. All this concern regarding the BMP is related to the efficient use of materials, which, as well as energy efficiency, contributes to the reduction of carbon emissions [60].

### 4.4. Building Environmental Certifications (BEC)

BEC is a systematic process of monitoring and evaluating whether a product, process, or service meets the pre-established requirements in technical standards and regulations while also having the lowest cost to society. Its objectives are to inform and protect the consumer, promote fair competition, encourage continuous quality improvement, facilitate international trade, and strengthen the internal market [61,62].

In the 1990s, the United States, Canada, and some European countries felt the need to create a system that would assess the environmental performance of their buildings. Thus, environmental certifications for buildings emerged, such as BREEAM (Building Research Environmental Assessment Method) in the United Kingdom, LEED (Leadership in Energy

and Environmental Design) in the United States, and BEPAC (Building Environmental Performance Assessment Criteria) in Canada. With this, they also thought about combating false "ecological buildings", which are propagated by some companies as a marketing strategy but which do not deliver significant benefits [63].

These certifications have evolved and been adopted in several other countries, even in those with very different realities. Currently, countries interested in improving the environmental performance of their buildings are creating their own certifications that are adapted to their local scenarios. Thus, each certification builds its values and emphasizes the sustainability criteria they deem most relevant [62].

The circular construction project covers the environment and mental and technical aspects together with governmental and behavioral dimensions. These are best developed through organizational tools, such as construction certification schemes [64].

Issues regarding climate change benefit from the promotion of renewable sources used to generate electricity and hot water as a way of mitigating the pollution derived from the burning of fossil fuels such as oil, coal, and natural gas. The search for energy efficiency in buildings through simple and innovative strategies, such as energy simulations, measurements, system commissioning, efficient design and construction, use of renewable and clean sources of energy generated on- or off-site, and use of equipment and efficient systems, are measures encouraged by environmental certifications [61–63].

Other activities that emit a large amount of $CO_2$ and GHG, such as disposal of waste in landfills and deforestation, are combated by certifications with the correct management of construction waste and the use of certified wood that comes from fast-growing, legalized, exotic, or native species. Greer et al. [65] state that certifications encourage the use of materials with low environmental impact (recycled, regional, recyclable, reused, etc.) and these decrease the generation of waste, in addition to promoting conscious disposal, which helps to remove the volume of waste generated from landfills.

Understanding the connection between energy and water consumption and its effect on GHG emissions is crucial for analyzing the effectiveness of BEC systems and thus contributes to the improvement related to the CE. The study of Greer et al. [65] shows how the adoption of the LEED certificate, related to energy and water efficiency, can contribute to reverse aspects of climate change. The study focuses on LEED v4, Building Design and Construction—(BD + C), which applies to the construction of new buildings. It investigates how some factors contribute to the reduction of GHG emissions; among the factors analyzed are water efficiency through reduction of indoor water use, reduction of water use in external environments, and energy efficiency through energy performance optimization. Energy use is the largest contributor to GHG emissions in the US, and energy efficiency is an important path to the CE. There is an incentive for greater energy efficiency and the use of alternative energy sources when adopting environmental certifications (especially LEED).

Likewise, reductions in water use in the buildings sector can be an important source of reductions in GHG emission because of the GHGs embedded in the energy spent to obtain, deliver, treat, and store water. Water systems demand energy both outside the building (for purchase, treatment, and transportation) and inside the building (for heating and, sometimes, on-site treatment). Household water use generally induces the demand for wastewater treatment, which varies in energy intensity based on the level of treatment required [66].

Based on this, BECs can contribute to a more circular process, especially in terms of reductions in resource consumption during building's operation (mainly energy, water, and the related GHG emissions), and the specification of materials with a higher content of recycled materials, as well as materials that can be reused or recycled in their end of life stage. In the last few years, as alternatives for gaining credit, some of these certifications, such as LEED and HQE, have started to use the LCA by requiring environmental products declarations (EPDs).

*4.5. Waste Management Plans (WMP)*

A WMP is a record of waste data that is generated based on the evolution of the construction (by construction stage, for example, foundation, structure, etc.; type of construction technologies used, for example, conventional vs. industrialized; the number of workers) to serve as a benchmark and a database for future projects [67,68].

The European Union's approach to waste management is currently based on two main pillars: on the one hand, a structure that favors waste prevention over reuse, followed by recycling, energy recovery, and finally, disposal, and on the other hand, the CE package that was adopted by the European Commission in 2015 advocates an economic system that prevents waste from being placed in landfills and maintains all material flowing in the economy through reuse, redesign, material recovery, or energy recovery. Given this, two main elements are introduced: the ban on landfills for certain types of waste and specific collection and recycling targets for the various existing wastes [69]. Reduction, reuse, and recycling, in this order of importance, are priorities in the waste management hierarchy [70]. Reduction refers to waste prevention, which Hutner et al. [71] define as any measure taken before a resource crosses the waste limit. They further argue that, despite their priority, prevention activities have been hesitant so far.

The same notions were presented in a comprehensive review by Ghisellini et al. [72], which claims that the principle of reduction remains one of the most poorly discussed topics in the scientific literature regarding the CE, and that the emphasis should shift to the use of smarter products. Reuse and recycling are the second and third priorities in the waste management hierarchy. Reuse is a preferable principle in general and is considered an effective way to reduce the volume of waste [44]. Theoretically, reuse overlaps with the reduction priority while encouraging users not to discard a product in the post-use phase. The first barrier to promoting reuse is the product design, an example of which is lithium-ion batteries that have low potential for reuse due to uncertainty in battery chemistry [73].

There is a need for CE-based waste management for valuable waste products in regions lacking infrastructure, capital, and tools to build a circular industrial economy. Despite these restrictions, WMP solutions must rely on the involvement of local consumers, and encourage them to assume ownership of their waste management rather than relying on government and industry assistance [70].

As a strategic model, WMPs have the aspects of reduction, reuse, and recycling in a planned way, thus facilitating the application of the CE model in the construction sector. Integrated waste management based on energy recovery from waste is vital to initiate general CE principles and industrial ecology concepts. A strategy based on mixed waste incineration and gas recovery from landfills in order to generate electricity provides global environmental benefits for sustainable development [74].

It is also possible to use WMPs as a means of mitigating climate change through material recovery, and various materials used in the construction sector, such as aluminum, iron, copper, paper, and plastic can be recovered. This recovery makes it unnecessary to manufacture new materials. It saves natural resources and reduces industrial GHG emissions into the atmosphere. This strategy is crucial for high-energy-intensive products, as in the case of some metals, e.g., aluminum. In another direction, we have the recycling of construction and demolition waste (CDW) as aggregates for cement-based materials to minimize the environmental impacts of landfill disposal and overcome the lack of natural resources [75].

Thus, as a tool for a CE, waste management is a strategic model that benefits by reducing, reusing, recovering, and recycling materials that are used in the construction sector after the end of their service life. In addition to this, WMPs allow us to quantify the minimization in relation to waste generated during construction.

Buyle et al. [76] studied the environmental impacts of seven alternative wall assemblies with five different types of end of life scenarios in Belgium: (i) current practice (incineration with energy recovery and recycling), (ii) maximized energy recovery, (iii) improved

recycling, (iv) optimized recycling (much higher recycling rates and off-site reuse), and (v) reuse in the same building without any additional treatment. In this case, emissions savings were 14% when current end of life waste management practices were adopted.

Although the WMP is a very important tool for waste sorting, identification, and adequate end of life use, we did not find an expressive amount of research on this topic in the literature, especially any linked with the CE. A possible explanation for this may be because it is a tool that is required in a lot of legislation and construction practice codes. In other words, it is already a tool used in market practice and in a mature way, and does not receive attention from researchers. One way to improve WMPs is to consider the application of BIM and BMP as complementary tools to be implemented during the waste management that happens during the building's construction and demolition/deconstruction.

### 4.6. Augmented Reality (AR) and Virtual Reality (VR)

Augmented reality (AR) is an extension of the real world in the form of computer-generated visual information in 3D models [77]. In the AEC industry, AR and VR are used for better visualization of projects integrated with BIM-like structural analysis for better understanding in the scientific community [78].

Virtual reality (VR) is a technology that creates virtual environments and replaces the user's perception of the surrounding environment with a virtual environment using HMDs, glasses, or other multi-display setups. Delgado et al. [79] categorized the use of AR and VR into six use-cases: stakeholder engagement, design support, design review, construction support, operation and management support, and training. VR has been adopted more than AR, and stakeholder engagement is the most adopted use-case. However, the AEC industry has implemented these tools to a lesser extent. For the future research agenda, three topics were placed as goals: defining engineering-grade devices, improving workflow and data analysis, and developing new technology resources in the area.

How can AR and VR technology help in the mitigation of climate change? New research from Stanford University examines how AR can change people's behavior in a way that can truly challenge the way people communicate. The simulation was made "by wearing goggles that layer computer-generated content onto real-world environments" and concluded that simple moves, such as the way you walk or the way you turn your head, are influenced by AR. This technological tool can be used in many situations, for example, for having business meetings, hence the cost of transportation and the GHG emissions that would be generated are avoided.

In the world of technology, Apple announced the update of its tablet in 2020 with the inclusion of the LiDAR sensor. This sensor uses infrared light to calculate the distance between objects and provide depth-associating AR; therefore, it is a tool that can be especially useful in the AEC industry. As an example, it facilitates the measurement of elements and the surveying of the terrain, which can improve the productivity of these tasks.

## 5. Use of the Tools in a Circular Building Environment

Based on the literature that was evaluated, we developed a flowchart (Figure 11) that shows where in the life cycle of buildings the tools can be applied and how they can contribute to climate change mitigation, and consider two types of design: conventional and disassembly (DfD). Additionally, the chosen tools are correlated with the CE strategies (Figure 12) and building's stakeholders (Figure 13).

Based on how the tool contributes to each phase, it is possible to link each tool throughout the building's life cycle to an action recommended by the CE. It can be noted that the BIM and LCA are the most applicable. Through design for disassembly (DfD), within this built environment, there are possibilities beyond recycling. DfD enables the future disassembly (or deconstruction) of buildings and the reuse and remanufacturing of construction components that contribute to reducing the use of natural resources and energy and, thus, the reduction of climate change.

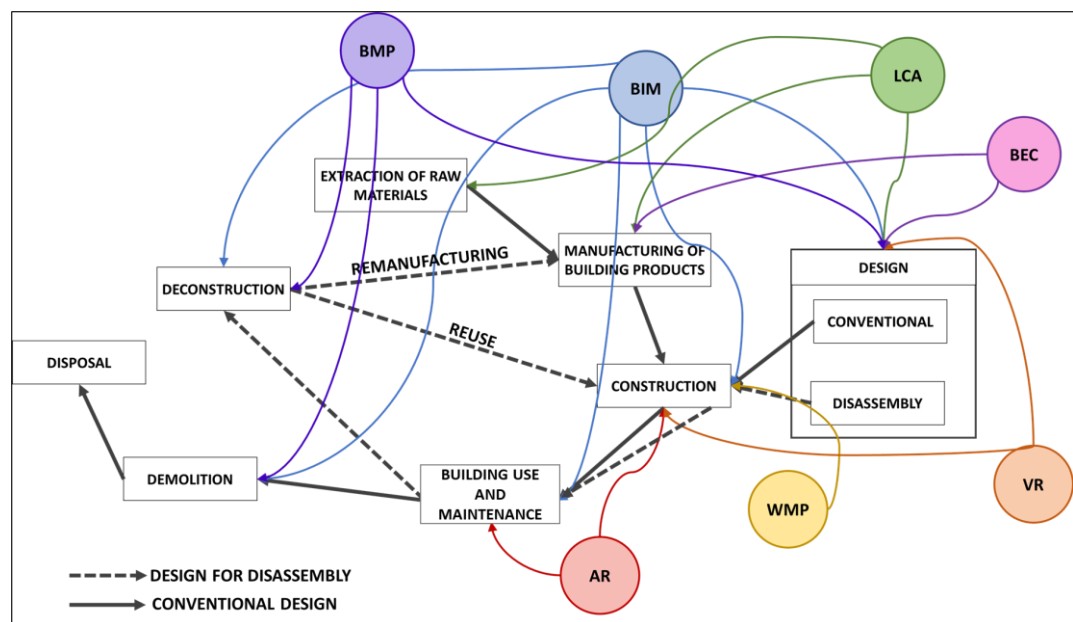

**Figure 11.** Use of the tools throughout the life cycle of buildings. BIM—building information modeling. LCA—life cycle assessment. BEC—building environment certification. AR—augmented reality. VR—virtual reality. WMP—waste management plan. BMP—building materials passports. Based on Cruz Rio et al. [31].

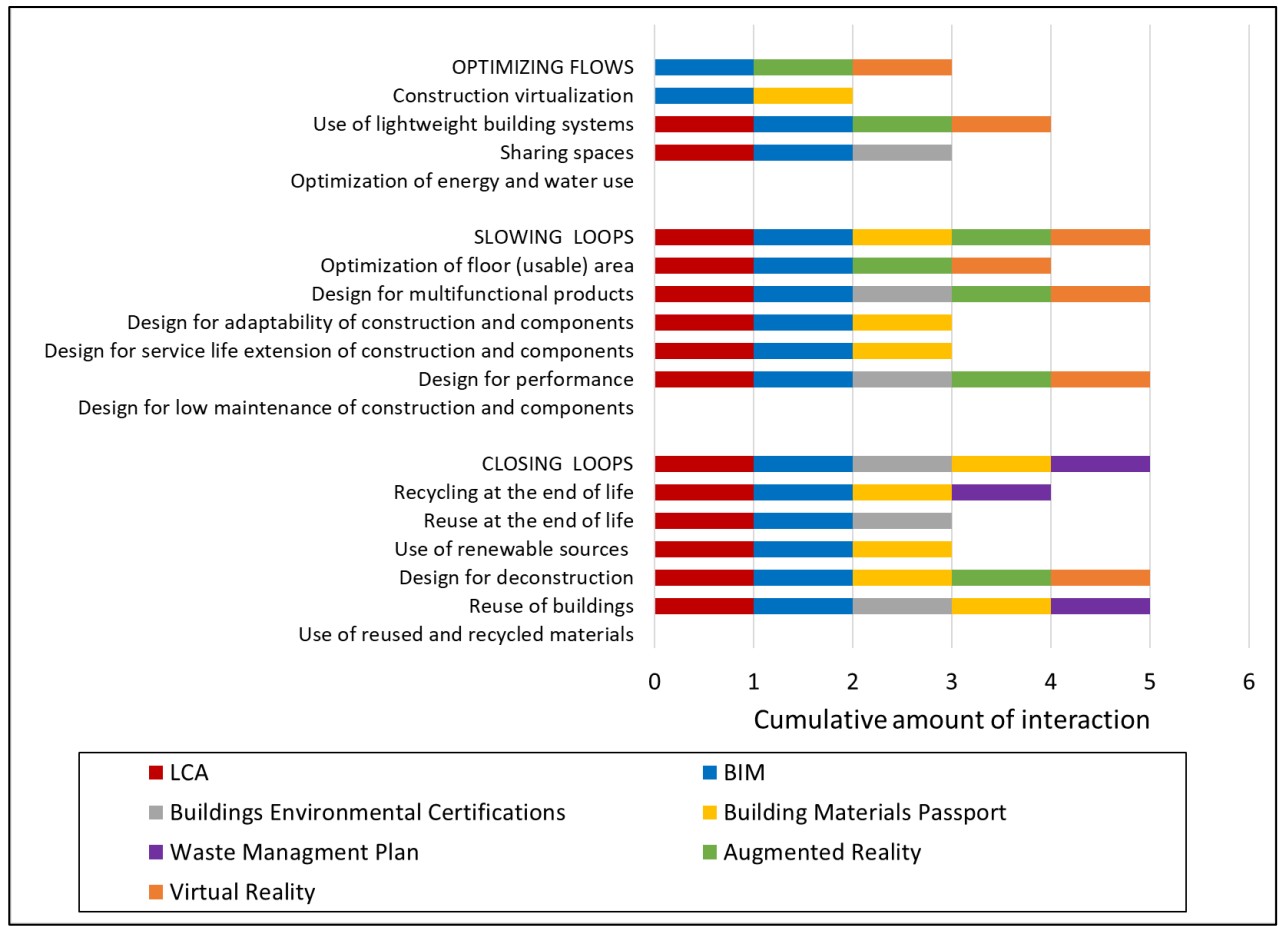

**Figure 12.** Use of tools according to CE strategies.

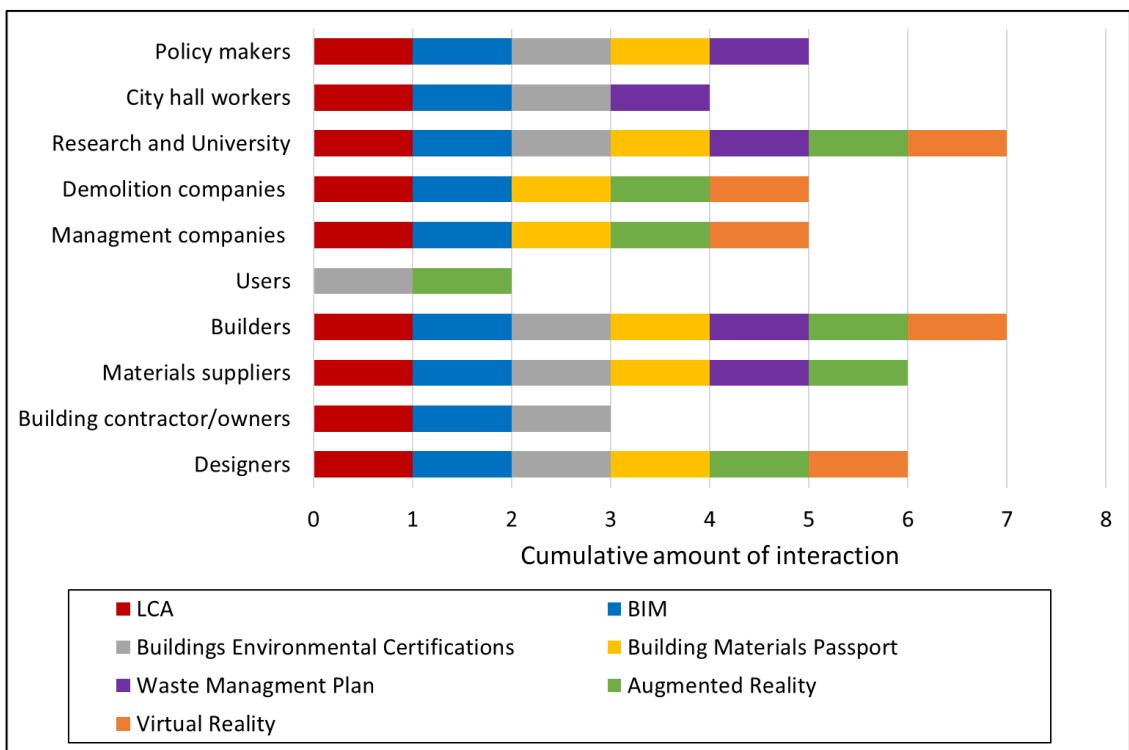

**Figure 13.** Use of tools according to the building's stakeholders.

Considering the building's stakeholders, the constructors, researchers, and designers can be the main users and, consequently, those that most benefit from the use of the evaluated tools. The LCA and BIM, once again, showed themselves to be the most applicable. However, each tool that is applied to CE strategies brings benefits and contributions to one or more stages of the building's life cycle, whether in the conventional building design process or in the disassembly (or deconstruction) project and can reduce GHG emissions, as can be observed in the following definitions:

(a)  LCA: Enables the quantification of benefits when using reused and recycled materials, reusing buildings, adopting the practice of DfD, using renewable energies, recycling and reusing materials, and building elements at the end of life stage. With LCA, it is possible to transform all options and strategies into GHG emissions reduction and measure the increase or decrease according to different strategies and scenarios.

(b)  BIM: Facilitates the design process by adopting CE strategies through automation and can be easily integrated with LCA, BMP, and BEC, and with AR and VR for modeling virtual products. It also facilitates optimizing the use of energy and water simulations, and consequently, the related GHG emissions, which, in turn, ensures the better environmental and thermo-energetic performance of buildings. BIM is a tool that can be present in all stages of the building's life cycle (project, material production, construction, use, maintenance, and end of life).

(c)  BEC: Encourages the use of recycled materials and their reuse in projects, in addition to requiring the adoption of CE strategies that contribute to the mitigation of climate change. It also encourages the use of renewable resources such as wood and bamboo that can absorb $CO_2$, the main GHG. It encourages functional projects and the use of materials that have easy maintenance and higher quality. It also seeks to encourage the rational use of construction materials and systems with less energy and water expenditure, less waste generation, and consequently, the related GHG emissions.

(d)  BMP: Its use has significant advantages in the design and deconstruction phases because it provides valuable information about the materials and wastes. Thus, it facilitates the deconstruction process and the destination of materials and construction elements by

identifying which materials have the potential for reuse and recycling. This information, which is provided from the materials, especially at the end of the building's life, is particularly important in order to estimate and reduce GHG emissions.

(e) WMP: Its use can be highlighted in the construction and end of life phases by making it possible to manage and quantify the waste generation, and it permits the selection of the best end-of-life option for waste. The data generated with this tool enable the quantification of GHG emissions. The sorting process can facilitate the reuse and recycling and, consequently, the reduction in related GHG emissions.

(f) AR and VR: These two can indicate elements of the building and construction in layers that facilitate the deconstruction process. The use of virtual models facilitates the evaluation of different strategies in the design phase, enables the choice of products to be used in each stage, and finally, allows the use of virtual models—mainly in the design, construction, and maintenance stages of the building. If it is possible to exchange a physical product for a virtual one, the GHG emissions during the life cycle of the physical product are avoided.

Based on the literature, we believe there is a trend towards the integration of the tools that use BMP with the tools that use BIM and LCA, as observed in Honic et al. [10]. In order to have a complete sustainability evaluation, with the evolution of the economic and social dimensions of LCA, the assessment will be more systemic. A scheme for the integration of the evaluated tools is proposed in Figure 14.

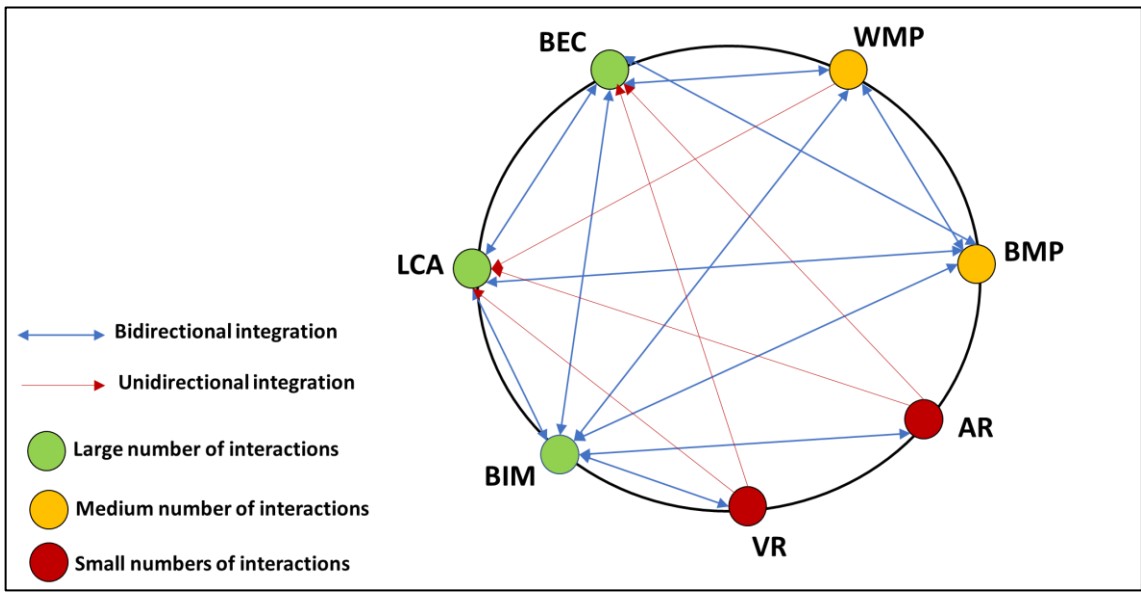

**Figure 14.** Integration of tools. LCA—life cycle assessment. BEC—building environment certification. AR—augmented reality. VR—virtual reality. WMP—waste management plan. BMP—building materials passports.

We can see that the tools can be classified mainly in three groups, according to the number of interactions (in green, yellow, and red colors): (1) strategic approach (LCA, BIM, and BEC); (2) strategic and visual approach (BMP and WMP); and (3) technological and execution approach (AR and VR). The first group allows a more systemic and holistic intervention related to general management strategies on a macro level. The second one has a similar action but on an intermediate level. The third one is used for the implementation of certain actions on a micro level.

LCA presents interactions with all other tools and shows its potential for adaptability and as a facilitator when choosing CE strategies for quantifying and mitigating GHG emissions. The BIM tool also has many interactions and the fact that all of its integrations are bidirectional is highlighted. BIM delivers accurate quantitative information from the

design phase to the end of the building's service life, thus enabling the application of other tools to enhance the measures, which can contribute to mitigating climate change.

The evaluated ICT tools, AR and VR, proved to have potential; however, so far, they have not been used (in the evaluated literature) in the context of sustainability and climate change mitigation (they have small numbers of interactions). We expect this research and integration between tools to happen in the near future, as long as this technology evolves and new sensors and software are made available for their best use in the sector. As shown in Figure 14, they can be linked mainly with LCA, BEC in a unilateral way, and BIM in a bilateral way.

We expect that shortly, all of these tools will be able to be used together, in a complementary way, with all the benefits for the production of circular economy buildings, though with a smaller amount of GHG life cycle emissions.

## 6. Recommendations for the Improvement of Tools

Based on the scientific evidence and the main objective of this research, which is the use of CE strategies and principles for mitigating climate change by reducing GHG emissions, we developed some recommendations for the improvement of the chosen tools (Table 2). These suggestions can be useful for the researcher, developers, and other stakeholders that use these tools.

**Table 2.** Recommendations for the improvement of tools in the context of the circular economy and climate change mitigation.

| Tools | Recommendations |
|---|---|
| Life cycle assessment (LCA) | -Use in preliminary design stages linked with other tools. Specifically, for BEC in the use of EPDs. |
| | -Link the different aspects of sustainability (environmental, economic, and social). |
| | -Make the benefits related to avoided impacts mandatory due to recovery of waste or closed-loop end of life scenarios (reuse, recycling, or burning with energy recovery). |
| | -Make the quantification of biogenic $CO_2$ mandatory for bio-based materials in order to account for the benefits related to the use of these materials for climate change mitigation. |
| | -Define clear and standardized rules to avoid problems such as double counting of benefits, frontiers of the second-life system, etc. Similar to PEF (product environmental footprint) in the European Union. |
| Building information modeling (BIM) | -Use in preliminary design stages linked with other tools. |
| | Development of a library of materials with information related to the level of circularity of the product (content of recyclable materials, potential for reuse or recycling at the end of life, etc.) |
| | -BIM software developers should create specific plug-ins related to waste management, circular product evaluation, design for disassembly (DfD), and the possibility of creating a building materials passport (BMP). |
| | -The BIM should be extended to an assessment tool beyond building boundaries, including the scale of neighborhoods and even cities (city information modeling). It is important to associate it with other tools such as the geographic information system (GIS). |
| Building environmental certifications (BEC) | -Create scores for other CE-related strategies that have not yet been considered (multifunctional projects, shared and collaborative projects, design for disassembly (DfD), end of life reuse and recycling, presence of materials passports, etc.) |

It is important to say that the gaps and recommendations observed in this research are not the only ones. Those that were listed were based on the papers evaluated in the SLR and deserve special attention in order for us to have more possibilities for scientific and technological progress and innovation in the context of CE.

## 7. Conclusions

Via a systematic literature review, this study evaluated how different tools are used to support decision making and thereby select circular economy (CE) strategies that can contribute to mitigating climate change in terms of GHG emissions reduction in the architecture, engineering, and construction (AEC) industry. The following tools were assessed: (1) life cycle assessment (LCA), (2) building information modeling (BIM), (3) building environmental certifications (BEC), (4) building materials passports (BMP), (5) waste management plans (WMP), (6) augmented reality (AR), and (7) virtual reality (VR).

A gap in scientific knowledge was observed concerning the use of tools focused on mitigating climate change. LCA and BIM are the main tools used in the evaluated papers, while the other ones, especially AR and VR are the least used. It is possible to see some integration between them, especially LCA, BIM, and BMP, and we propose other kinds of integration in the future, considering the improvement and actualization of the tools. LCA presents interactions with all the other tools evaluated and shows its potential for adaptability and as a facilitator when choosing CE strategies; it is also essential for quantifying GHG emissions.

It was observed that most of the reviewed studies focused on strategies that were classified as closing loops, mainly related to recycling and reuse at the end of life and the use of recycled materials. The relationship between the use of these tools and the building's stakeholders was also evaluated and it was noted that constructors, researchers, and designers can be the main users and, consequently, those that most benefit from the use of these tools.

Furthermore, the classification of the tools into the following three groups is suggested: (1) a strategic approach (LCA, BIM, and BEC); (2) a strategic and visual approach (BMP and WMP); and (3) a technological and execution approach (AR and VR). This classification can help researchers and users to obtain an easier interpretation and understanding in terms of how these tools can be used to reduce GHG emissions in a circular building environment.

Finally, some suggestions are proposed for the improvement of the evaluated tools in order to address the question of a CE aligned with climate change mitigation in the AEC industry. These suggestions should be a point of interest for future studies and should bring insights into new technology and innovation for the AEC industry.

**Author Contributions:** Conceptualization, L.R.C. and M.T.M.C.; methodology, L.R.C. and M.T.M.C.; formal analysis, L.R.C., M.V.S., V.P.S. and M.T.M.C.; investigation, L.R.C., M.V.S., V.P.S. and M.T.M.C.; data curation, L.R.C., M.V.S., V.P.S. and M.T.M.C.; writing—original draft preparation, L.R.C., M.V.S. and V.P.S.; writing—review and editing, L.R.C., M.V.S., V.P.S., M.T.M.C. and R.D.T.F.; visualization, L.R.C. and M.T.M.C.; supervision, R.D.T.F.; project administration, L.R.C., M.T.M.C. and R.D.T.F. All authors have read and agreed to the published version of the manuscript.

**Funding:** This research received no external funding.

**Institutional Review Board Statement:** Not applicable.

**Informed Consent Statement:** Not applicable.

**Conflicts of Interest:** The authors declare no conflict of interest.

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
