# Peer review of "How Different Tools Contribute to Climate Change Mitigation in a Circular Building Environment?—A Systematic Literature Review"

_sustainability, doi:10.3390/su14073759_

Round 1

Reviewer 1 Report

Dear Author,

Overall it is a very fine works with sustematic methodology.

I just thought the last part of the paper, i.e. the Reference can start from the fresh page, and not at the end of the page.

Thank you and best wishes.

Reviewer 1

Author Response

Sustainability

MDPI

Prof. Dr. Marc A. Rosen

Editor-In-Chief

Sybil Han

Assistant Editor

Subject: Revision of the Manuscript sustainability-1507215

REVIEWER 1

  1. I just thought the last part of the paper, i.e. the Reference can start from the fresh page, and not at the end of the page.

Thank you for this comment. We have changed it.

Prof. Lucas Rosse Caldas, PhD

Civil Engineering Program- PEC/COPPE/UFRJ

Postgraduate Program in Architecture - PROARQ/UFRJ

Federal University of Rio de Janeiro (UFRJ) – Brazil

Reviewer 2 Report

How can the use of different tools contribute to climate change 2 mitigation in a circular building environment? A systematic 3 literature review

The paper investigated seven tools related to CO2 emmission calculation,  in construction and carry out a Systematic Literature Review on the theme to analyze these tools.

  • The topic might be shortened.
  • It is not clear why and how these tools are selected? Besides they are not in line with eac other nor wok as a complement series.
    • Line 88.

“The following tools (management and technological) were chosen in our research: (1) Life cycle assessment (LCA), (2) Building Information Modeling (BIM), (3) Building Envi- ronmental Certifications (BEC), (4) Building Material Passports (BMP), (5) Waste Manage- ment Plans (WMP), (6) Augmented Reality (AR) and (7) Virtual Reality (VR).”

  • The introduction is too sketchy, and not yet developed. The research gap is not clear. There are a lot of previous research papers, which could not be regret in last five years and former.
  • It is too noisy and indistinct. It is not even clear how many papaers are investigated as the mentioned numbers are in conflict with each other; Nor the rank of the journals/papers are presented.
  • Line 137.

“Further reading of the introduction and conclusion was carried out, and only 12 papers were selected as relevant. The snowball approach added 18 new articles to the analysis, some previously known by the authors, totaling 30 publications.”

  • The numbers mentioned in Fig 1 is not consistent with in the text.
  • The methodology is not clear. Although the methods of data gathering is a part of methodology, analysis and extraction are to be mentioned, while the introduction and categorization of the data are more necessary in review papers, which is not performed well in the paper.
  • Once more, there is not clear if the selected terms are homological or equal ? the answer to this uestion change the deduction of the survey (and its parts of course !) ( fig 4 and Line 224….230)

“The 224 number of publications found is described in Figure 5. It can be seen that the number of 225 some tools, such as BIM and LCA, were much more explored, with a total of 7,719 publi- 226 cations on the two themes (including duplicate articles). However, the other construction 227 tools were much less researched, finding a gap in the area that can be further explored, 228 especially when it comes to BEC, BMP, and WMP.”

  • The authors investigated 30 papers and find that the main theme aspect of “5.40” papers are in optimized flow. !!!!!!! (30*18%=5.40) :

  • Line 235.

“An analysis of the CE main categories was made based on the 30 articles that met all the content requirements in the methodology. It was noted that 68.8% of the articles have as main theme aspects of closing loops, 13.3 % in slowing loops, and 18% in optimized flow, according to Figure 6. The detailing of CE strategies evaluated is presented in Figure 7.”

  • Part 4, “Qualitative Analysis” should be restructured in introduction and be discussed and analysed in part 4 ! the structure of the text is a kind of fact accumulation which should be changed to a scientific clear well structured data analysis/ review paper.
  • The figures (fig 12 , 13) are not clear. What is the meaning of 1…6 and 1…..8 in horizontal axis?
  • The results are not clear nor practical, and some are more negligible to be mentioned.
  • “Result and Discussion” are the main parts of a “review paper”, which should be based on the method and research gap extracted from introduction. None of these characteristics are reflected in the submitted paper.
  • Many recent papers (2020,2021 and 2022) are not pointed, nor referenced !!

Writing a review paper is not simple. A detaile/holistic comprehensive scientific archive mixed with the ability to data management and deduction is the start point, and a lot of experience and practice is needed to provide an acceptable review paper.

Author Response

Sustainability

MDPI

Prof. Dr. Marc A. Rosen

Editor-In-Chief

Sybil Han

Assistant Editor

Subject: Revision of the Manuscript sustainability-1507215

REVIEWER 2

  1. The topic might be shortened. It is not clear why and how these tools are selected? Besides they are not in line with each other nor wok as a complement series.

Thank you for this comment. We have added details explaining how these tools were selected. 

  1. The introduction is too sketchy, and not yet developed. The research gap is not clear. There are a lot of previous research papers, which could not be regret in last five years and former.

Thank you for this comment. We have improved the introduction section and clarified the research gap. We have also added previous research papers about this theme. 

  1. It is too noisy and indistinct. It is not even clear how many papers are investigated as the mentioned numbers are in conflict with each other;

Thank you for this observation. We have corrected this.

  1. The numbers mentioned in Fig 1 is not consistent with in the text.

Thank you for this observation. We have corrected this.

  1. Part 4, “Qualitative Analysis” should be restructured in introduction and be discussed and analysed in part 4! the structure of the text is a kind of fact accumulation which should be changed to a scientific clear well structured data analysis/ review paper.

Thank you for this comment. We have reviewed the whole section, in terms of content and text. However, since the other reviewers did not criticize this Section structure, we prefer to maintain the same structure. 

  1. The figures (fig 12 , 13) are not clear. What is the meaning of 1…6 and 1…..8 in horizontal axis?

Thank you for this comment. We have added a title in a horizontal axis explaining this.  

  1. The results are not clear nor practical, and some are more negligible to be mentioned.

Thank you for this comment. We have reviewed the Results section.   

  1. Many recent papers (2020,2021 and 2022) are not pointed, nor referenced !!

Thank you for this comment. We have added some recent papers to correct this.  

Prof. Lucas Rosse Caldas, PhD

Civil Engineering Program- PEC/COPPE/UFRJ

Postgraduate Program in Architecture - PROARQ/UFRJ

Federal University of Rio de Janeiro (UFRJ) – Brazil

Reviewer 3 Report

This paper review on how can the use of different tools contribute to climate change  mitigation in a circular building environment? A systematic  literature review . Generally, this research work is worthy of affirmation. However, some deficiencies need to be improved before considering it acceptance.

General comments:

-       Written English must be improved in essentially all the sections of the paper. It is responsibility of the authors’ to provide a readable text that is, within a reason, grammatically correct. It may seem unfair, but failure to converge to a readable document is grounds for rejection.

Specific comments

Abstract section:

  • Authors need to rewrite the abstract. Highlight the scientific value added by your paper in your abstract. The abstract should clearly describe the core of the problem you are addressing, what you did, found and recommend to the readers. It will help a prospective reader of the abstract to decide if they wish to read the entire article.
  • The abstract should include the most important results
  • Abstract and Keywords: Life Cycle Assessment, Building Information Modeling replace by LCA; BIM
  • The keywords should include the most important results

Introduction section :

  • Authors need to add a significance section of this study with reference to past studies for highlighting the novelty of this study.

Methodology section

The authors should provide better explanation for the readers what was the difference between existing literature Eberhardt et al. [17] and the present study.

Quantitative Analysis section

Figure 3. Good results but need to explain and discuss in details

Figure 7a-c. Good results but need to explain and discuss in details

  • Conclusions and Final Remarks are also clumsy and have to be entirely rewritten; furthermore, for sound, comprehensive and meaningful conclusions the variability of results has to be taken into consideration.
  • Add some of the important findings to conclusion

Author Response

Sustainability

MDPI

Prof. Dr. Marc A. Rosen

Editor-In-Chief

Sybil Han

Assistant Editor

Subject: Revision of the Manuscript sustainability-1507215

REVIEWER 3

1. Written English must be improved in essentially all the sections of the paper. It is responsibility of the authors’ to provide a readable text that is, within a reason, grammatically correct. It may seem unfair, but failure to converge to a readable document is grounds for rejection.

Thank you for this comment. The English has been corrected by a professional reviewer who is a native speaker. A revision certificate is enclosed.  

2. Authors need to rewrite the abstract. Highlight the scientific value added by your paper in your abstract. The abstract should clearly describe the core of the problem you are addressing, what you did, found and recommend to the readers. It will help a prospective reader of the abstract to decide if they wish to read the entire article. The abstract should include the most important results.

Thank you for this comment. The abstract was rewritten and now highlights the scientific contribution of the paper, the description of the core problem and the most important results.

3. Abstract and Keywords: Life Cycle Assessment, Building Information Modeling replace by LCA; BIM.

Thank you for this comment. We have corrected this in the text.

4. Authors need to add a significance section of this study with reference to past studies for highlighting the novelty of this study.

Thank you for this comment. We have added two paragraphs on previous studies about the theme and have highlighted the novelty of our study.

.

5. The authors should provide better explanation for the readers what was the difference between existing literature Eberhardt et al. [17] and the present study.

Thank you for this comment. We have added a paragraph explaining the differences between Eberhardt et al. [17] and the present study. The reference for the citation Eberhardt et al. [17] was corrected to Charlotte et al.

.6. Figure 3. Good results but need to explain and discuss in details.

Thank you for this comment. We have added a paragraph explaining and discussing Figure 3 in greater detail.

7. Figure 7a-c. Good results but need to explain and discuss in details.

Thank you for this comment. We have added two paragraphs explaining and discussing Figure 7a-c in greater detail.

8. Conclusions and Final Remarks are also clumsy and have to be entirely rewritten; furthermore, for sound, comprehensive and meaningful conclusions the variability of results has to be taken into consideration. Add some of the important findings to conclusion.

Thank you for this comment. The Conclusion and Final Remarks have been entirely rewritten. We have also added the main important findings of the study.

Prof. Lucas Rosse Caldas, PhD

Civil Engineering Program- PEC/COPPE/UFRJ

Postgraduate Program in Architecture - PROARQ/UFRJ

Federal University of Rio de Janeiro (UFRJ) – Brazil
